# Optimisation of Cable Dome Structure Design for Progressive Collapse Resistance

**Lian-Meng Chen** [1,*], **Kai-Yu Huang** [1], **Yi-Jie Liu** [1], **Yi-Hong Zeng** [1], **Ze-Bin Li** [1], **Yi-Yi Zhou** [2] **and Shi-Lin Dong** [3]

1 College of Civil Engineering and Architecture, Wenzhou University, Wenzhou 325035, China
2 College of Civil Engineering and Architecture, Changzhou Institute of Technology, Changzhou 213002, China
3 Space Structures Research Center, Zhejiang University, Hangzhou 310027, China
* Correspondence: 00151034@wzu.edu.cn; Tel.: +86-13957790090

**Abstract:** Since the literature lacks an effective analysis method of collapse mechanisms and optimisation design theory for progressive collapse resistance of cable dome structure, a structural performance-based optimisation approach was proposed to improve the progressive collapse resistance for cable dome structures in this study. First, the dynamic response and collapse model of a cable dome structure were analysed after its members were removed using Ansys LS-DYNA and the full dynamic equivalent load-based instantaneous unloading method. Second, the importance coefficients of the members were calculated to determine the contribution of each member to the progressive collapse resistance of the structure. Finally, a stepwise optimisation solution was proposed by integrating a global optimisation model, which uses the mean of the importance coefficients of all members as the optimisation index, with a local optimisation model, which minimises the maximum member importance coefficient. The results indicated that different members exhibited varying levels of importance in the progressive collapse resistance of the structure, with the inner and outer hoop cables demonstrating the highest levels of importance, followed by the inner upper string of the tension hoop. The other members had low levels of importance. Compared with the cable dome structure based on the Geiger topology, the cable dome structure based on the Levy topology was more resistant to progressive collapse; such resistance decreased as the number of cable-truss frames decreased. Additionally, the local optimisation approach based on the genetic algorithm reduced the maximum member importance coefficient (i.e., that of the outer hoop cable) by 60.26%.

**Keywords:** cable dome structures; progressive collapse; member importance analysis; design optimisation



## 1. Introduction

According to Fuller's idea of tensegrity, cable dome structures are a type of flexible structure formed through tension with tension cables and struts as the basic elements [1]. Given the high tensile strength of these cables and the application of prestress for structure stiffness adjustment, these structures are widely used in construction and the commonly used types are the Geiger cable dome [2] and Levy cable dome [3]. Economic development and technological advancements have contributed to the development of cable dome structures towards longer spans, more complex forms, and more novel materials. Meanwhile, the fact that numerous incidents of building collapse have been reported as a result of extreme loading environments, such as explosions and extreme weather conditions [4–6], has merited research into the analysis and evaluation of the performance (e.g., resistance to progressive collapse) of cable dome structures and the optimisation of their design.

Current studies on cable dome optimisation have focussed on the optimisation of prestress, element cross-sectional area, structural shape, and structural topology. Yuan [7]; Chen and Dong [8]; Zhang [9]; and Liang, Dong, and Miao [10] proposed prestress optimisation design models with the optimisation goals of minimising the level of global prestress applied to the cable dome structure, minimising the maximum initial internal force of members (typically the initial internal force of the outer hoop cable), minimising the initial strain

of the structure, minimising the mass of the structure, and minimising the reaction force of the support. Zhang and Sun [11]; Yuan [7]; Wu [12]; and Liu [13] optimised the element cross-sectional areas of various cable dome structure components by minimising the cost of construction, minimising the weight, and optimising the stiffness. Zheng [14] increased the evenness of internal force distribution in the net structure of the main cables of the Five-hundred-meter Aperture Spherical radio Telescope by optimising the element cross section through the method of constant tensile force replacement. Xie and Steven [15] proposed a progressive structural optimisation approach in which the influence of each component on the global performance of structures is evaluated to eliminate elements with the least contributions, thereby optimising the topology and shape of structures. Ma et al. [16], Zhao et al. [17], and Shan [18] developed lattice structure topology optimisation models based on structure robustness and the solid isotropic material with penalisation method, a variable density method. Liu [19] established a shape optimisation model for cable dome structures by using the multipopulation genetic algorithm and particle swarm algorithm by adopting control element length as the independent variable, and by employing minimum structural response in a loaded structure as an optimisation indicator. As for optimisation algorithm, some interesting and novel algorithms have been developed and applied, such as surrogate-assisted stochastic optimisation inversion algorithm [20], shrimp and goby association search algorithm [21], velocity pausing particle swarm optimisation [22], and grey wolf optimiser algorithm [23].

Overall, the current approaches of prestress optimisation for cable dome structures have mostly been focussed on minimising the prestress level or minimising the maximum member prestress in the load case. Optimised designs of element cross section, structural shape, and structural topology have been based on the optimisation objectives of minimising the structure mass and achieving optimal stiffness. In practice, a cable dome structure is lightweight, typically with a steel quantity less than 30 kg/m$^2$; therefore, the benefit of optimising the mass of a structure is limited. In addition, optimised designs based on structure performance, particularly resistance to progressive collapse in an extremely complex load case, are uncommon.

The first progressive collapse, a concept proposed in the field of civil engineering, was observed in 1968 at Ronan Point, a 22-storey block of flats in London, UK, which progressively collapsed because of a gas explosion [24]. Following the 9–11 terrorist attack in 2001, the complete collapse of the World Trade Center in the United States drew global research attention towards the concepts of progressive collapse and structural robustness in buildings [25]. Corley et al. [26] analysed the Murrah Federal Building incident and argued that redundant load paths are essential for preventing asymmetric building collapse. According to Tsopelas and Husain [27], Izzuddin [28], Dusenberry and Juneja [29], Houghton and Karns [30], and Mendis and Ngo [31], increasing the strength and ductility of key building components is a crucial measure against progressive structure collapse. Ma, Chen, and He [32] introduced the concept of structure alternative defensive ability, which refers to the ability of structures to automatically adjust their internal forces when certain components fail because of unexpected events or excessive local overloading. This ability allows structures to halt the progression of damage, thereby preventing their global damage and progressive collapse. Wang et al. [33], Hui et al. [34], Yuan et al. [35], and Zhang et al. [36] simulated cable breakage by using the element birth and death technique, instantaneous removal of components, and instantaneous loading to analyse the local failure of various cable dome structures as well as the internal force change in members and nodal displacement response following breakage. Jiang [37], Cai et al. [38], and Zeng et al. [39] analysed the local failures, dynamic structural response, and collapse of various string structures, such as the beam string structure, the cable–arch structure, and the truss string structure. Countries and regions such as North America [40,41], Europe [42,43], and China [44,45] have incorporated requirements for the prevention of structural collapse and the improvement of structural robustness into their design guidelines, standards, and regulations.

In summary, the literature on the progressive collapse of structures has predominantly been focused on frame structure systems and has rarely investigated long-span structures. This is because traditional long-span structures, such as space trusses and grid shells, are generally believed to have a relatively high degree of static indeterminacy, and the failure of a single member does not substantially undermine the bearing-capacity reserve of the global structure. However, unlike traditional long-span space trusses or grid shells, cable dome structures have low redundancy and high sensitivity to accidental disturbances, including construction errors, and are thus prone to collapse when they are overloaded or subjected to accidental disturbances [4–6].

To address the problems associated with cable dome structures, this study analysed typical models of cable dome structure dynamic response and collapse resulting from local member failure by using Ansys LS-DYNA and the full dynamic equivalent load-based instantaneous unloading method. Structural displacement ratios were also compared before and after the removal of components to define the member importance coefficients in order to determine the importance of each component in the progressive collapse resistance of structures. Finally, to optimise the performance of structures in resisting progressive collapse, a stepwise optimisation design solution based on a combination of global and local optimisation was proposed.

## 2. Dynamic Response and Model Analysis of Progressive Collapse

### 2.1. Cable Dome Structure Case

The structure analysed in this study was the roof of Yi Jin Huo Luo Qi sports centre in Inner Mongolia, which is constructed using a Geiger cable dome with a span of 71.2 m and a rise of 5.5 m, with the hoops divided into 20 equal segments. Each cable-truss frame consists of 13 members arranged axially symmetrically to the centre: an outer diagonal cable (DC1), a middle diagonal cable (DC2), an inner diagonal cable (DC3), an outer ridge cable (RC1), a middle ridge cable (RC2), an inner ridge cable (RC3), an outer strut (OS), a middle strut (MS), an inner strut (IS), an outer hoop cable (OHC), an inner hoop cable (IHC), an inner upper string of tension hoop (IUS), and an inner lower string of tension hoop (ILS). The design load is 0.4 kN/m$^2$, and a hinged support is fixed to the compression hoop beams. Figure 1 depicts the plan and cross section of the roof, and Table 1 illustrates the cross-sectional parameters and initial prestress of each component. The elastic moduli of the tension cables and struts are 160 GPa and 206 GPa, respectively.

### 2.2. Cable Elements, Strut Elements, and Equivalent Force Model

Ansys LS-DYNA was used to simulate cable and strut elements by selecting LINK167 and LINK160, and the offset amount was defined to apply prestress. The equations used are as follows:

$$F = K \times \max\{\Delta L, 0.0\} \tag{1}$$

$$K = EA/(L_0 - \text{offset}) \tag{2}$$

where $\Delta L$ is the amount of change in strut length, $L_0$ is the initial strut length, E is the elastic modulus, A is the cross-sectional area, and offset is the amount of offset. For the LINK160 strut element, a bilinear dynamic material model was used, with the failure strain set at 0.01. In other words, any strut elements with a strain level higher than 0.01 were removed from the structure [34].

Analyses were performed using the full dynamic equivalent load-based instantaneous unloading method, which involves removing one component of the structure and replacing it with an equivalent force before unloading the equivalent force. The replacement time, duration, and unloading time of the equivalent force were set as twice, 20 times, and 1/10, respectively, of the remaining natural period of vibration of the structure (which was set to 2 s in this study) [34,40]. Figure 2 depicts the entire process of the equivalent force action.

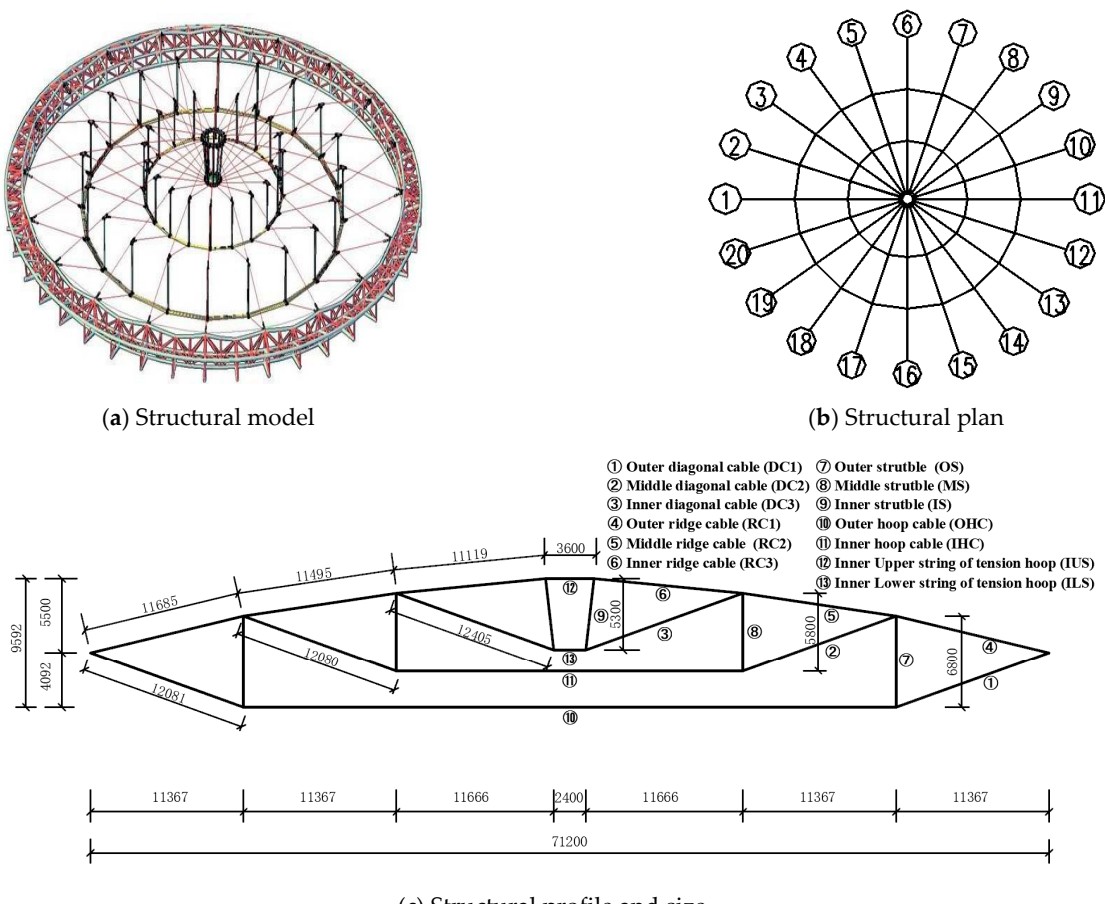

(**a**) Structural model　　　　　　　　　　　　　　　　　　(**b**) Structural plan

① Outer diagonal cable (DC1)　⑦ Outer strutble (OS)
② Middle diagonal cable (DC2)　⑧ Middle strutble (MS)
③ Inner diagonal cable (DC3)　⑨ Inner strutble (IS)
④ Outer ridge cable (RC1)　⑩ Outer hoop cable (OHC)
⑤ Middle ridge cable (RC2)　⑪ Inner hoop cable (IHC)
⑥ Inner ridge cable (RC3)　⑫ Inner Upper string of tension hoop (IUS)
　　　　　　　　　　　　　⑬ Inner Lower string of tension hoop (ILS)

(**c**) Structural profile and size

**Figure 1.** Investigated cable dome structure.

**Table 1.** Parameters and initial pre-stress of initial model elements.

| Member | DC1 | DC2 | DC3 | RC1 | RC2 | RC3 | OS | MS | IS | OHC | IHC | IUS | ILS |
|---|---|---|---|---|---|---|---|---|---|---|---|---|---|
| Area (mm$^2$) | 2490 | 853 | 605 | 1840 | 1360 | 853 | 7800 | 4670 | 4670 | 7470 | 3320 | 3320 | 3320 |
| Prestress (kN) | 466.6 | 208 | 105.9 | 682.2 | 473.1 | 370 | −158 | −70.4 | −36.2 | 1403.2 | 625.7 | 1190.1 | 305.3 |

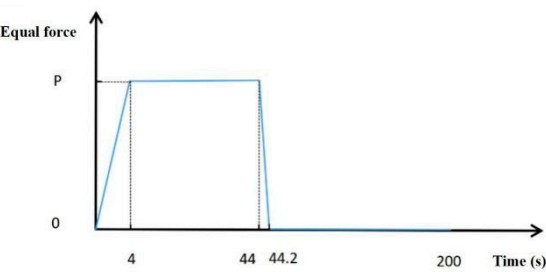

**Figure 2.** Whole progress of the equivalent force.

*2.3. Dynamic Response and Collapse Model Analysis for the Entire Process of Structural Collapse*

Because of space limitations, analyses were performed only for the displacement response, internal force response, energy response, and collapse model of the structure after the OHC was removed.

(1)　Displacement response: Figure 3a,b illustrate the vertical displacement response when the equivalent force began to be unloaded (t = 44 s) and when the structure reached its final state with damping (t = 200 s), respectively. The results indicated the following. First, the node displacement was consistent with the initial structure once

the structure reached a balance after the cable was replaced with an equivalent force. Second, Nodes 2 and 9, which were connected to the OHC, rapidly moved towards the sides after the OHC was removed (equivalent force). Their movement resulted in varying degrees of horizontal displacement in other OHC elements and all members connected to the OHC. Third, the failure of the OHC and the resultant reduction in the OS support led to a large vertical displacement in the upper and lower nodes of all OSs. Additionally, Nodes 3 and 2 in the OS, which were directly connected to the failed member, were vertically displaced by 5.03 m and 2.49 m, respectively. The displacement of other OS nodes in the cable-truss frames negatively correlated with their distance from the failed member.

(2) Energy response: Figure 4, which depicts the change in internal kinetic energy over a period of 200 s, reveals the following results: First, the structure exhibited a certain kinetic energy response and remained at an imbalanced state after the failed member was replaced by an equivalent force, and the kinetic energy gradually decreased with damping and reached 0 at 44 s. Second, when equivalent force unloading began at t = 44 s, the structure transitioned again from a balanced state to an imbalanced state, with the redistribution of the internal force that contributed to the generation of a relatively large kinetic energy, which gradually decreased with damping. At approximately 75 s, the structure approached a balanced state, with a kinetic energy of 0. The text following an equation need not be a new paragraph. Please punctuate equations as regular text.

(3) Internal force response: Figure 5, which depicts the time-dependent internal force change in members in the cable-truss frame adjoining the left-hand side of the failed OHC, reveals the following results. First, the internal forces of the RC1, OHC, and OS, all of which were directly connected to Node 2, reached 0 after the OHC was removed. These members continued to exert an effect on the structure after the redistribution of the structure's internal forces. The OHC experienced the largest loss of prestress, with a reduction of 90%. Second, the internal forces of other members exhibited varying degrees of reduction, with the largest reduction (90%) observed in the DC1.

(4) Collapse model: After the OHC was removed, the structure produced a substantial dynamic response. Almost all components exhibited a displacement greater than 1/50 of the span. When the structure reached its final balanced state, all prestress was lost, and the structure completely collapsed. The largest displacement was observed at Node 17 (a node on the OS), which exhibited a vertical displacement of 5.58 m.

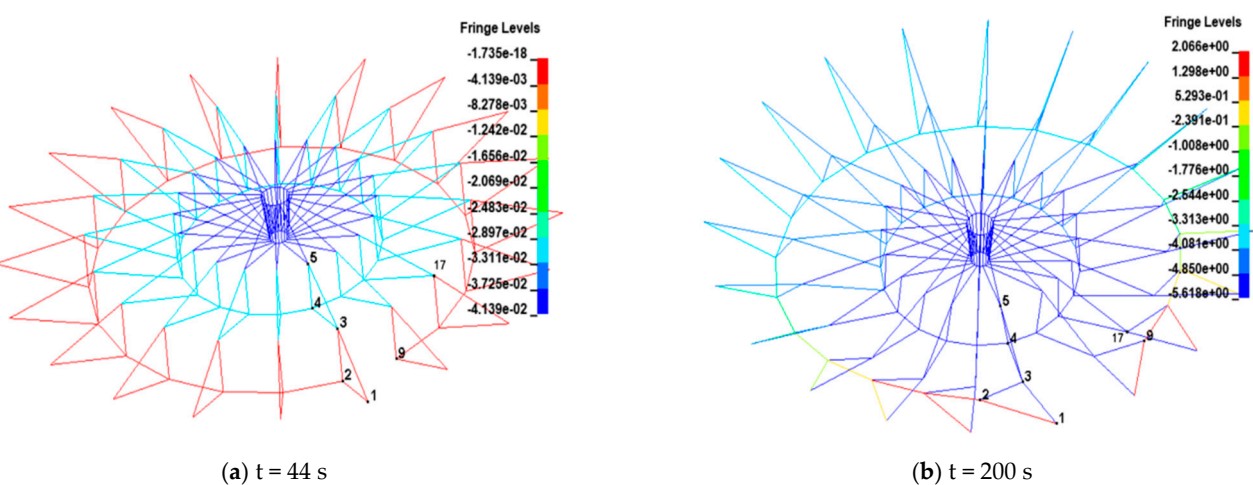

(**a**) t = 44 s          (**b**) t = 200 s

**Figure 3.** Displacement response.

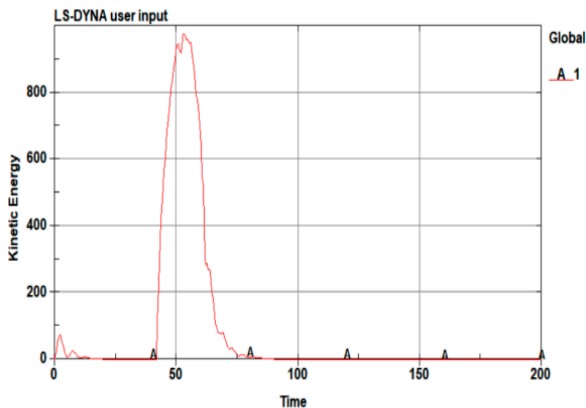

**Figure 4.** Time–history of energy response.

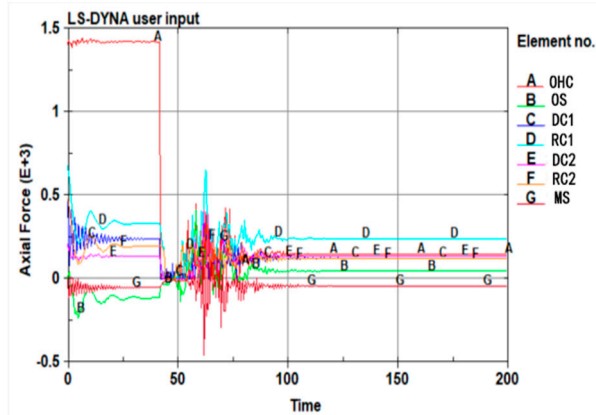

**Figure 5.** Time–history of internal force response.

Table 2 presents the analysis results of the dynamic response and collapse model of the structure after each member was removed. The results indicated that the removal of various members resulted in different dynamic responses and collapse models. Particularly, the removal of the IHC and OHC resulted in relatively large collapsed areas and vertical displacement, followed by those observed after the removal of the IUS. The removal of the DC1, DC2, DC3, RC1, RC2, RC3, OS, MS, IS, and ILS yielded the smallest collapsed areas and vertical displacement. Hence, each factor contributed differently to the structure's progressive collapse.

The correlation between each member's importance level and collapse model was further explored using the US UFC4-023-03 criteria [40] for the progressive collapse of cable dome structures. These criteria are as follows: First, a progressive collapse is said to have occurred in a cable dome if the maximum vertical displacement of a node is greater than 1/50 of the span and if the area of failure reaches 30% of the dome's total planar area. Second, a local progressive collapse is said to have occurred in a cable dome if the maximum vertical displacement of a node is greater than 1/50 of the span and if the area of failure is less than 30% of the dome's total planar area. Third, a progressive collapse is said to have not occurred if the maximum vertical displacement of a node is less than 1/50 of the span or if the maximum vertical displacement of the node is greater than 1/50 of the span, with an area of failure less than 15% of the dome's total planar area.

In this study, these criteria were used to categorise cable dome collapse after the removal of members into three models: a progressive collapse model, a partial collapse model, and a nonprogressive collapse model. According to the model used, the corresponding removed component was then identified as a key component, an important component, or a common component (Table 2).

**Table 2.** Types of collapse induced by the removal of different types of cable–strut members and the importance categories of these members.

| Member | Description of Collapse | Collapse Models | Importance Coefficient | Important Properties |
|---|---|---|---|---|
| DC1 | 10% of the collapsed area; the maximum vertical displacement was 0.64 m at Node 3 (upper node of the outer strut) | Non-progressive collapse | 0.01 | Common component |
| DC2 | 0% of the collapsed area; the maximum vertical displacement was 0.36 m at Node 5 (upper node of the middle strut) | Non-progressive collapse | 0.0049 | Common component |
| DC3 | 0% of the collapsed area; the maximum vertical displacement was 0.10 m at Node 7 (upper node of the inner strut) | Non-progressive collapse | 0.0009 | Common component |
| RC1 | 10% of the collapsed area; the maximum vertical displacement was 9.93 m at Node 3 (upper node of the outer strut) | Non-progressive collapse | 0.023 | Common component |
| RC2 | 4.6% of the collapsed area; the maximum vertical displacement was 8.30 m at Node 5 (upper node of the middle strut) | Non-progressive collapse | 0.01 | Common component |
| RC3 | 0% of the collapsed area; the maximum vertical displacement was 0.10 m at Node 7 (upper node of the inner strut) | Non-progressive collapse | 0.0015 | Common component |
| OS | 0% of the collapsed area; the maximum vertical displacement was 1.42 m at Node 3 (upper node of the outer strut) | Non-progressive collapse | 0.0061 | Common component |
| MS | 0% of the collapsed area; the maximum vertical displacement was 1.10 m at Node 5 (upper node of the middle strut) | Non-progressive collapse | 0.002 | Common component |
| IS | 0% of the collapsed area; the maximum vertical displacement was 0.10 m at Node 7 (upper node of the inner strut) | Non-progressive collapse | 0.000074 | Common component |
| OHC | 100% of the collapsed area; the maximum vertical displacement was 5.58 m at Node 136 (upper node of the outer strut) | Progressive collapse | 0.54 | Key component |
| IHC | 47% of the collapsed area; the maximum vertical displacement was 3 m at Node 126 (upper node of the inner strut) | Progressive collapse | 0.23 | Key component |
| IUS | 16.6% of the collapsed area; the maximum vertical displacement was 1.56 m at Node 3 (upper node of the outer strut) | Partial progressive collapse | 0.15 | Important component |
| ILS | 16.6% of the collapsed area; the maximum vertical displacement was 0.756 m at Node 7 (upper node of the inner strut) | Non-progressive collapse | 0.033 | Common component |

## 3. Member Importance Analysis

On the basis of the aforementioned US criteria [40], which employ structural displacement and area of collapse as indices of the collapse model, this section further compares structural displacement before and after a component is removed to determine its importance level by using the following equation:

$$\rho_i = \frac{\|s_{1i} - s_0\|}{\|s_0\|} \tag{3}$$

where $\|\ \|$ denotes the Euclidean norm, and $s_0$ and $s_{1i}$ are the displacement vectors of the loaded structure before and after component i is removed, respectively. A small $\rho_i$ indicates a small displacement response of the structure after component i is removed and hence a low-importance component. By contrast, a large $\rho_i$ indicates a high-importance component.

The importance coefficient of each member was analysed in a stepwise manner and then normalised using the following equation:

$$\beta_j = \rho_j / \sum_{i=1}^{13} \rho_i \tag{4}$$

Figure 6, which presents the importance coefficients of the 13 members, reveals the following results. First, the members varied in terms of their importance coefficients, with the OHC having the largest importance coefficient, followed by the IHC. Removing these two members caused the structure to produce a large dynamic response, and thus they were classified as key components. Second, the IUS, which had the third-largest importance coefficient, was classified as an important component, suggesting that its removal would

considerably affect the structure. Third, the remaining members were all categorised as common components because of their low importance coefficients, meaning that their removal would not result in (local) progressive collapse. Fourth, the importance levels of different members were as follows: hoop cables > ridge cables > diagonal cables > struts.

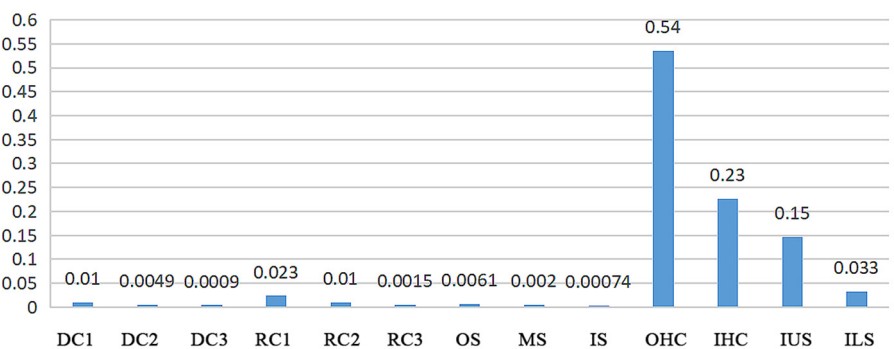

**Figure 6.** Importance coefficient of different members.

## 4. Optimised Design

Different topological forms of cable dome structures vary in terms of their resistance to progressive collapse. To increase the progressive collapse resistance of the cable dome included in this study, a stepwise optimisation approach integrating global and local optimisation was proposed.

### 4.1. Globally Optimised Design

The importance coefficient of each member was factored into the evaluation of the structure's global resistance to progressive collapse. Specifically, the mean of the importance coefficients of all members ($\overline{\rho}$) was used as an index to globally optimise the cable dome design for optimal resistance to progressive collapse and thus explore the optimal topological form of the structure:

$$\overline{\rho} = \frac{1}{n}\sum_{i=1}^{n}\rho_i = \frac{1}{n}\sum_{i=1}^{n}\frac{s_{1i} - s_0}{s_0}, i = 1, 2, 3, \cdots, n \tag{5}$$

A small $\overline{\rho}$ indicates that the removal of all members has a minimal global effect on the structure, suggesting high resistance to progressive collapse.

(1) Topological relationship: Figure 7a–d illustrate four distinct cable dome topological forms. Dome 1 is the original structure model. Dome 2 is the original model with the OHC's topological relationship changed from a Geiger arrangement to a Levy arrangement. Dome 3 is a Dome 2 model with the middle hoop cable's topological relationship changed from a Geiger arrangement to a Levy arrangement. Dome 4 is a Dome 3 model with the IHC's topological relationship changed from a Geiger arrangement to a Levy arrangement.

Figure 8, which depicts the global performance index values of the aforementioned topological forms, reveals the following results: First, the transition from a Geiger arrangement to a Levy arrangement of hoop cables from the OHC progressively inward to the IHC caused the global performance index ($\overline{\rho}$) to decrease from 1.04 in Dome 1 to 0.43 in Dome 4. This indicated that the global resistance to progressive collapse gradually increased as the topological forms changed from a Geiger arrangement to a Levy arrangement. Second, the structural redundancy increased upon the transition from a Geiger arrangement to a Levy arrangement and resulted in the global resistance to progressive collapse increased.

(2) Number of cable-truss frames: Figure 7e,f illustrate two distinct cable dome with different cable-truss frames. Domes 5 and 6 build on the original model with the original 20 cable-truss frames reduced to 12 and 16 frames, respectively, and the global performance index values of the aforementioned two domes were depicted in Figure 8. When the number of cable-truss frames was reduced from 20 in Dome 1 to 16 in Dome 6 and 12 in Dome 5,

the area affected by each cable-truss frame and each member increased with the global performance index increasing from 1.04 in Dome 1 to 1.20 in Dome 6 and 1.46 in Dome 5, indicating a gradual decrease in the structure's resistance to progressive collapse. On the contrary, the global performance index reduced, the area affected by each cable-truss frame and each member reduced, and the global resistance to progressive collapse increased with the increase in cable-truss frames.

(3) More discussions: First, the study analysed only the transition from a Geiger arrangement to a Levy arrangement and how the change in the number of cable-truss frames affected the structure's global resistance to progressive collapse. In general, the increase in structural redundancy, the decrease in the area affected by the members, and the decrease in the importance of each member all contributed to an increase in resistance. However, a more scientific and precise design for topological optimisation falls outside the scope of this study. Second, global optimisation based on changes in the structure's topology and number of cable-truss frames decreased the importance coefficient of the OHC (which had the largest importance coefficient of all members) from 0.5355 in Dome 1 to 0.5083 in Dome 4. Therefore, changing a structure's topological form and increasing its redundancy enhance its global resistance to progressive collapse and reduce the importance of its members.

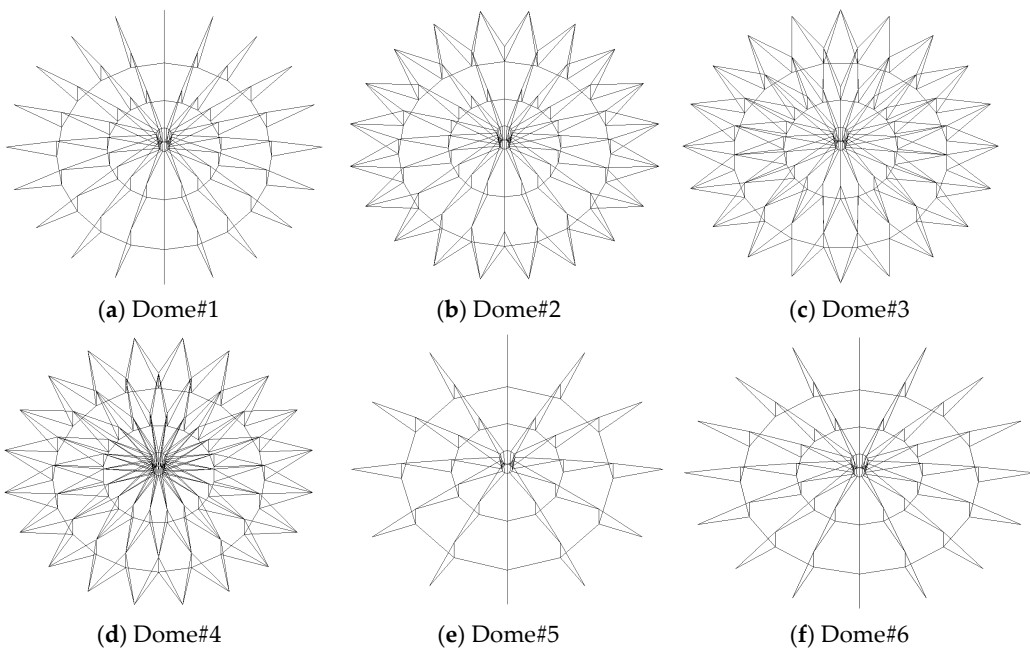

(**a**) Dome#1      (**b**) Dome#2      (**c**) Dome#3

(**d**) Dome#4      (**e**) Dome#5      (**f**) Dome#6

**Figure 7.** Different forms of cable dome structures.

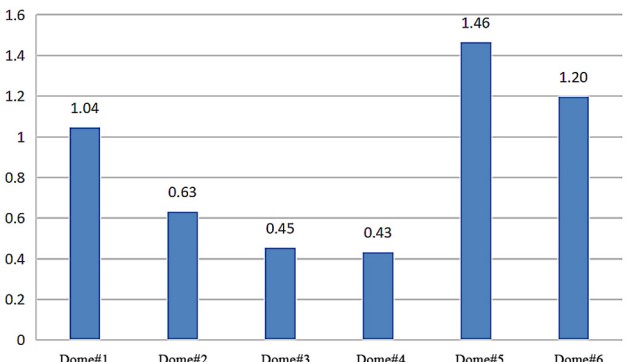

**Figure 8.** Global performance index under different forms.

### 4.2. Locally Optimised Design

Building upon the results of global optimisation, this subsection presents a local optimisation process for Dome 4, that is, the Levy cable dome structure that exhibited the highest global resistance to progressive collapse among all six designs. According to the member importance analysis results, the removal of a member that has a large importance coefficient can result in a high dynamic response in the structure and may cause progressive collapse. Therefore, in this study, an optimisation model minimising the maximum member importance coefficient ($\beta_{max}$) was developed. In the current structure, the optimisation target was the importance coefficient of the OHC (Member 10), namely $\min\beta_{10}$. Parameter analysis revealed that, for structure systems with fixed topological relationships, the shape parameters (e.g., hoop cable radius and strut length) more sensitively influenced the importance coefficients of the members compared with the design parameters (e.g., cross-sectional area and prestress level). Therefore, the following mathematical model for local optimisation was proposed:

$$\begin{cases} \min\beta_{10} = \rho_j / \sum_{j=1}^{13} \rho_j \\ X_{i,min} \leq X_i \leq X_{i,max} \end{cases} \tag{6}$$

where $X_i$ is the optimisation variable. As depicted in Figure 9, three combinations of shape parameters were used as the optimisation variables: (1) height differences of the support node (Node 1), with Node 3 on the OS ($S_1$) and Node 5 on the MS ($S_2$); (2) lengths of the OS ($H_1$) and MS ($H_2$); and (3) radii of the OHC ($R_1$) and IHC ($R_2$). In (6), $X_{i,max}$ and $X_{i,min}$ refer to the upper and lower limits, respectively, of the optimisation variable in question and are set as 120% and 80% of the variable's initial value, respectively (Table 3).

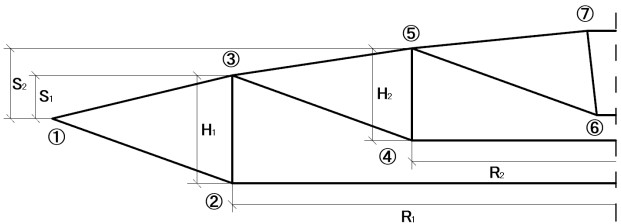

**Figure 9.** Optimisation variables.

**Table 3.** Control range of optimisation variables.

| Optimisation Variables | Initial Values | Lower Limits | Upper Limits | Optimisation Results |
|---|---|---|---|---|
| $S_1$ | 2.708 | $S_{1min} = 2.166$ | $S_{1max} = 3.249$ | 2.712 |
| $S_2$ | 4.418 | $S_{2min} = 3.534$ | $S_{2max} = 5.302$ | 4.745 |
| $H_1$ | 6.800 | $H_{1min} = 5.440$ | $H_{1max} = 8.160$ | 8.096 |
| $H_2$ | 5.800 | $H_{2min} = 4.640$ | $H_{2max} = 6.960$ | 6.878 |
| $R_1$ | 24.233 | $R_{1min} = 19.386$ | $R_{1max} = 29.080$ | 29.080 |
| $R_2$ | 12.866 | $R_{2min} = 10.293$ | $R_{2max} = 15.439$ | 13.575 |

An interactive genetic algorithm was used in MATLAB and Ansys LS-DYNA for design optimisation. The optimisation process was as follows: First, individuals with appropriate lengths were generated to form the initial population in MATLAB according to the solution preciseness and number of variables, and the population size was set to 20. Second, the basic information of each individual in the initial population in MATLAB was input into Ansys LS-DYNA to construct a basic structural model and calculate the prestress of each member through an iterative approach. Third, the fitness value was calculated according to the optimisation index in Ansys LS-DYNA, and the result was saved and then sent back to MATLAB. Fourth, the calculation results of each individual were summarized and analysed, the best individual was selected, and both its individual code and individual

solution were recorded. Fifth, a new code was generated through crossover and mutation on the basis of the code of the fittest individual. Here, the probabilities of crossover and mutation were set to 0.8 and 0.2, respectively. Sixth, the number of iterations (200 in this study) determined whether the optimisation process continued or stopped. In case the process continued, the optimisation process started over from the first step. However, in case the process stopped, the optimisation process proceeded to produce the final optimal solution.

Figure 10 illustrates the iterative optimisation of the importance coefficient of the OHC. First, the structural index $\beta_{10}$ decreased as the number of iterations increased, and it remained constant after the 82nd iteration. Second, the index $\beta_{10}$ decreased by 60.26% from 0.5083 to 0.2020 in the optimisation process. In the optimised design, the parameters were as follows: $S_1 = 2.712$, $S_2 = 4.745$, $H_1 = 8.096$, $H_2 = 6.878$, $R_1 = 29.080$, and $R_2 = 13.575$. Figure 11 depicts the model parameters before and after optimisation. Third, when optimisation was conducted separately with each of the three parameter combinations, the optimisation effectiveness (measured by the percentage of $\beta_{10}$ reduction) was greatest when the hoop cable radii were changed (53.73%), followed by when the strut lengths (10.35%) and then the height differences of the strut nodes (1.01%) were changed.

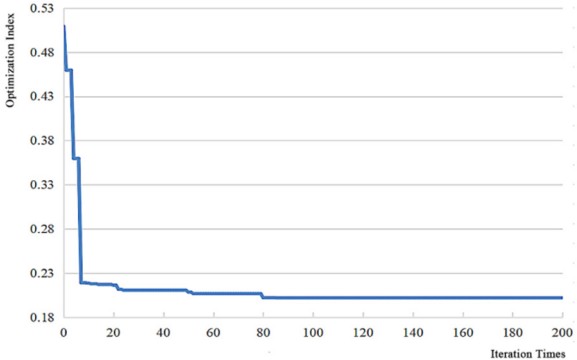

**Figure 10.** Optimisation iteration progress.

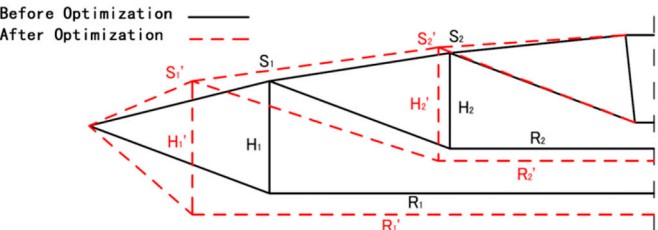

**Figure 11.** Initial shape and optimised shape.

## 5. Conclusions

In this study, since the literature lacks an effective analysis method of collapse mechanisms and optimisation design theory for progressive collapse resistance of cable dome structure, a structural performance-based optimisation approach was proposed to improve the progressive collapse resistance for cable dome structures. First, the dynamic response and collapse model of a cable dome structure were analysed after the failure of its members by using Ansys LS-DYNA and the full dynamic equivalent load-based instantaneous unloading method. The importance coefficients of the members were then calculated to determine each member's importance in the structure's resistance to progressive collapse. Finally, a stepwise optimisation model was developed by combining global and local optimisation. The results were as follows:

(1) The removal of different members resulted in a variety of dynamic responses and collapse models within the structure. For instance, the removal of the IHC and OHC resulted in the largest dynamic responses, with the largest importance coefficients,

and thus they were classified as key components. The removal of the IUS resulted in the third-largest dynamic response, with the third-largest importance coefficient, and thus it was classified as an important component. The other members were associated with relatively small dynamic responses, with low importance coefficients, and thus they were classified as common components. These results indicated that the importance of various members in the structure's resistance to progressive collapse varied.

(2) A global performance index was proposed to describe the global resistance to progressive collapse. The progressive collapse resistance of the cable dome structure decreased as the number of members decreased. The structure based on the Levy topology was more resistant to progressive collapse than that based on the Geiger topology.

(3) Local optimisation based on the genetic algorithm effectively reduced the maximum member importance coefficient (i.e., that of the OHC) by 60.26%. The optimisation process performed by changing the hoop cable radii was also much more effective than changing other shape parameters.

**Author Contributions:** Conceptualization, L.-M.C.; methodology, L.-M.C. and Y.-J.L.; software, Z.-B.L. and K.-Y.H.; validation, Y.-J.L., K.-Y.H. and Y.-H.Z.; resources, L.-M.C., Y.-Y.Z. and S.-L.D.; writing—original draft preparation, L.-M.C. and Y.-J.L.; writing—review and editing, L.-M.C. and K.-Y.H.; supervision, L.-M.C., Y.-Y.Z. and S.-L.D.; project administration, L.-M.C. All authors have read and agreed to the published version of the manuscript.

**Funding:** This study was supported by the National Natural Science Foundation of China (Grant No. 51578422, 51678082).

**Institutional Review Board Statement:** Not applicable.

**Informed Consent Statement:** Not applicable.

**Data Availability Statement:** Not applicable.

**Conflicts of Interest:** The authors declare no conflict of interest.

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
