# Peer review of "Optimisation of Cable Dome Structure Design for Progressive Collapse Resistance"

_applsci, doi:10.3390/app13042086_

Round 1

Reviewer 1 Report

The current study is worked on the optimization of cable dome structure considering dynamic behavior of the system. The LS-DYNA program is applied for the structural analysis. The reported results can be useful for the readers in the field of structural engineering. The article is recommended for publication after implementing the following comments:

Comments:

- Why the Authors uses the “dynamic analysis” in the work. Is it referred to the snap-through mechanism or any dynamic loads. It should be cleared in the manuscript.

- It is suitable that authors describe the different types of dome configurations and their specification before analysis section to provide more clear perspective for the readers.

- Although the LS-Dyna program is applied for the analysis, the main analysis concepts for this type of structures should be given for the readers.

- The collapse mechanisms in the dome structure should be described for readers (nodal instability, member instability and domino effect local/global collapse)

- To rise the quality of the work the self-citations should be reduced

Reviewer 2 Report

In this article, an alternative method to those that already exist is presented in order to improve the structures' resistance to progressive collapse (specifically for cable dome structures). Overall, the article is interesting and well organized. However, taking into consideration the following comments may help to improve some parts of this article:

Abstract Section:  Include a few sentences that provide an explanation as to why it is essential to apply the recommended methodology for enhancing the structural design of dome structures. What is the knowledge gap in the literature?

Introduction Section:  Add a paragraph or sentences that emphasizes what differentiates the methodology that is being proposed from past approaches as well as the knowledge gap that exists in the existing body of literature.

Conclusion Section: Briefly explain in a few sentences: What was the knowledge gap? How does this study fill this knowledge gap?

Some formatting issues. 

Reviewer 3 Report

The authors have presented a research paper entitled " Optimisation of Cable Dome Structure Design for Progressive Collapse Resistance". Overall, the paper is well-written and well-organized. Please address the following issues:

1.   The boundary conditions of the problem should present in detailed.

2.   The advantages and disadvantages of approach should be discussed.

3.   Optimal results are highly random, and hence how are authors concluding these results the best?

4.   Because of the important role of algorithm, the authors should add a paragraph to present the recent algorithms. The authors might be interested in the methods:

·     A surrogate-assisted stochastic optimization inversion algorithm: Parameter identification of dams. Advanced Engineering Informatics, 55, 101853.

·     A new metaheuristic algorithm: Shrimp and Goby association search algorithm and its application for damage identification in large-scale and complex structures. Advances in Engineering Software, 176, 103363.

·     Velocity pausing particle swarm optimization: a novel variant for global optimization. Neural Computing and Applications, 1-31.

·     A new movement strategy of grey wolf optimizer for optimization problems and structural damage identification. Advances in Engineering Software, 173, 103276.

5.   The authors should check typing errors throughout the whole paper. Many mathematical equations looks not professional in format. English style should also be improved.

Round 2

Reviewer 3 Report

I have no further comments.